# Q-MoE: Connector for MLLMs with Text-Driven Routing

## ABSTRACT

Multimodal Large Language Models (MLLMs) have showcased remarkable advances in handling various vision-language tasks. These models typically consist of a Large Language Model (LLM), a vision encoder and a connector structure, which is used to bridge the modality gap between vision and language. It is challenging for the connector to filter the right visual information for LLM according to the task in hand. Most of the previous connectors, such as light-weight projection and Q-former, treat visual information for diverse tasks uniformly, therefore lacking task-specific visual information extraction capabilities. To address the issue, this paper proposes Q-MoE, a query-based connector with Mixture-of-Experts (MoE) to extract task-specific information with text-driven routing. Furthermore, an optimal path based training strategy is also proposed to find an optimal expert combination. Extensive experiments on two popular open-source LLMs and several different visual-language tasks demonstrate the effectiveness of the Q-MoE connecter. We will open our codes upon publication.

## CCS CONCEPTS

• **Computing methodologies** → *Natural language processing*; **Computer vision**.

## KEYWORDS

Multimodal Large Language Model, Query-Based Connector, Mixture-of-Experts

## 1 INTRODUCTION

Visual-Language Multimodal Large Language Models (MLLMs)[2, 5, 10, 16, 21, 22, 27] exhibit significant potential in handling various visual-language tasks, such as Visual Question Answering(VQA)[12, 28, 32], Image Captioning[25, 35] and Referring Expression Comprehension(REC)[17].

Instead of starting from scratch, most of MLLMs are built on Large Language Models (LLMs) for inheriting the powerful capability of LLMs. A typical MLLM consists of a LLM, a vision encoder and a connector. The vision encoder, typically ViT[8], is used to encode visual inputs for MLLM. The connector maps visual codes to inputs of the LLM. Given a visual encoder, the connector solely determines what visual information will be transferred to the LLM. It is the bottleneck of visual information, and, therefore, crucial for the MLLM to accomplish various vision-language tasks. It is

*Conference'17, July 2017, Washington, DC, USA*
© 2024 Association for Computing Machinery.
ACM ISBN 978-x-xxxx-xxxx-x/YY/MM...$15.00
https://doi.org/10.1145/nnnnnnn.nnnnnnn

Figure 1: **Structure comparison between Q-MoE and the previous single expert connector.** Q-MoE includes Mixture of task experts and a text-guided router. The text embedding is used to select expert in each layer. Note that blue lines donate vision stream and black lines donate text stream.

challenging for the connector to filter the right visual information for LLM according to the task at hand.

Existing connectors mainly include two types of designs. One is a projection layer used in LLaVA[27], GIT[37], InfMLLM[41], FROMAGe[19], Shikra[4] and DreamLLM[7] etc, by which visual codes are directly mapped into the word embedding space by a trainable projection matrix. It is a lightweight structure with the merit of quick training. However, it is difficult for the projection itself to transfer task-specific visual information to the LLM; although instruction fine-tuning is helpful to alleviate the problem, an extra large number of visual instructions are needed. The other is query-based connector as used in BLIP2[21], InstructBLIP[5], RegionBLIP[42], MiniGPT4[43],mPLUG-owl[39],VistaLLM[30] and VPGTrans[40], etc. Q-former [21] is a typical query-based connector. It is a Transformer which introduces a set of learnable query vectors as image inputs to extract visual features from the visual encoder by cross-attention. Two-stage training makes the learnable query vectors in Q-Former learn to extract visual representation that is most informative of the text and tokens.

Textual information is introduced by cross-attending learnable query vectors in the Q-former structure. It brings an effective text-based visual information extraction. However, as pointed out in [21], the model does not achieve an improved VQA performance when providing the LLM with the in-context VQA example. We argue that one reason is that more accurate visual information for different tasks should be extracted. Inspired by the success of Mixture-of-Expert (MoE)[15] in the pre-training language model and LLM[20,31,9,29,44,11,24,38], we therefore try to employ the MoE structure into Q-former as the second stage extractor for more accurate visual information, where text embedding is used for

expert routing. It also enhances the model's capability of dealing with various different vision-language tasks with different experts. Furthermore, different from previous MoE training, we explore expert path based MoE training.

On the whole, this paper proposes **Q-MoE**, a new **Q**uery-based connector aiming to utilize the Mixture-of-Expert(**MoE**) to achieve more accurate text-driven visual information. Specifically, we build a MoE structure into Q-former as well as a novel router, Cross-Router, which routes different query tokens to their Task Experts (modelling by Feed-Forward Networks (FFNs)) by making use of cross attention between the text representation and output of each expert. Furthermore, an optimal expert path based training method, ExpertPath, is employed. Instead of selecting expert layer-wise, an optimal expert path across multiple layers is selected for training. It turns local training of experts in previous work into global optimization of experts. In this way, an expert is an expert path in our model. It is, therefore, called optimal expert path based training, which helps to maximize the effectiveness of expert combination.

We validate the effectiveness of Q-MoE structure with optimal path based training strategy under the multi-task fine-tuning setting, as well as zero-shot setting in various vision language tasks, including general Visual Question Answering [12, 14], knowledge-based VQA [28, 32], image captioning[25], and text-based image captioning tasks(TextCaps)[35]. Our model achieves remarkable improvements compared with the previous Q-former structure.

The contributions of our work are summarized as follows:

- We propose a novel connector **Q-MoE**, a new structure of enabling Q-former with MoE with the text-driven router. This structure enables second stage visual information extraction for more accurate visual information for the text.
- We further develop the optimal expert path based training strategy, ExpertPath, for MoE, which treats an expert as a path consisting of several experts in different layers. This approach upgrades the selection of layer-wise local expert FFNs to the selection of optimal global FFN paths.
- Experiments conducted under multi-task settings as well as zero-shot setting demonstrate the remarkable proficiency of our model in various vision language tasks.

## 2 RELATED WORK

### 2.1 Multimodal Large Language Models (MLLMs)

To inherit the powerful capability of LLMs, most of the MLLMs are built on the foundation of LLMs, with a visual encoder and a connector to bridge the modality gap between vision and language.

There are two lines to develop the connector among the current works. The first line simply uses a projection module to align representation of each modality[27, 37,41, 19, 4, 7]. The projection module can be a one or two-layer MLP. It is a lightweight structure and offers the advantage of rapid training. Although works following this line can project representations effectively, redundancy and low calculation efficiency still exist. Notably, as the resolution of images increases, the token numbers of one image and the associated computation cost would escalate dramatically. Furthermore, it is

difficult for the projection to transfer the task-specific visual information to the LLM, causing a performance decline in downstream tasks.

Differing from the first line, works in the second line could decrease the representation cost of a single image. Flamingo[2] introduces the Perceiver Resampler, which sorts to a set of learnable latent queries to interact with variably sized visual features outputted by the vision encoder through layers or cross-attention and FFNs. Q-former is proposed by BLIP2[21]. It is initialized by a BERT-base[6] model with an inserted cross-attention layer every two layers, where n learnable tokens are taken as queries and visual hidden states as keys and values. Q-former is broadly used to multiple other MLLMs [5, 42, 43, 39, 30, 40, 16]. InstructBLIP[5] leverages Q-former by concatenating query tokens and instructions to extract the instruction-aware visual features. LLaMA-VID [22] abstracts one frame in a video to 2 tokens, one is a text-guided context token from a context-attention module, while the other is a visually-enriched content token. These query-based connectors fuse text-related visual information into query tokens and input them into the LLMs. However, they treat visual information for diverse tasks uniformly, lacking of task-specific extraction capabilities. To address such issues, we design a Mixture of Experts(MoE) structure in Q-former along with text-driven routing to extract more task-relevant visual information.

### 2.2 Mixture-of-Experts (MoE)

Mixture-of-Experts (MoE)[15] is a hybrid model consisting of multiple sub-models, known as experts, combining the outputs of experts via a router in an input-dependent way. Each of these experts has its unique set of trainable weights, enabling them to generate distinct representations for each input based on contextual information.

The MoE has been thoroughly investigated in the field of Computer Vision[31], Natural Language Processing [9, 44] and Multimodal Learning[29]. The core component within the MoE architecture is the router, which determines the extent to which experts are differentiated. Works related to router design can be categorized into hard router [3, 23, 34] and soft router[11, 20, 24, 29]. In the hard router mode, each expert is typically pre-defined as a specific pattern without the need for router learning. Soft routers facilitate the dynamic distribution of data among diverse experts, empowering each to concentrate on its specialized area and attain model sparsity.

The Mixture-of-Experts structure has achieved remarkable success in pre-training language models and Large Language Models. Gshard[20] scales Transformer models in LLMs by replacing alternate Transformers' FeedForward layers with Sparsely-Gated MoE layers. Furthermore, findings from [33] suggest that the LLMs enhanced with MoE exhibit more significant benefits from instruction tuning when compared to dense LLMs. Recently in the multimodal domain,[24] transforms LLaVA[27] structure to the MoE type, and its 3B sparse-activated model achieves comparable performance with 7B models. [11] proposes MoCLE, the Mixture of Cluster-conditional LoRA Experts architecture, in which the routing of

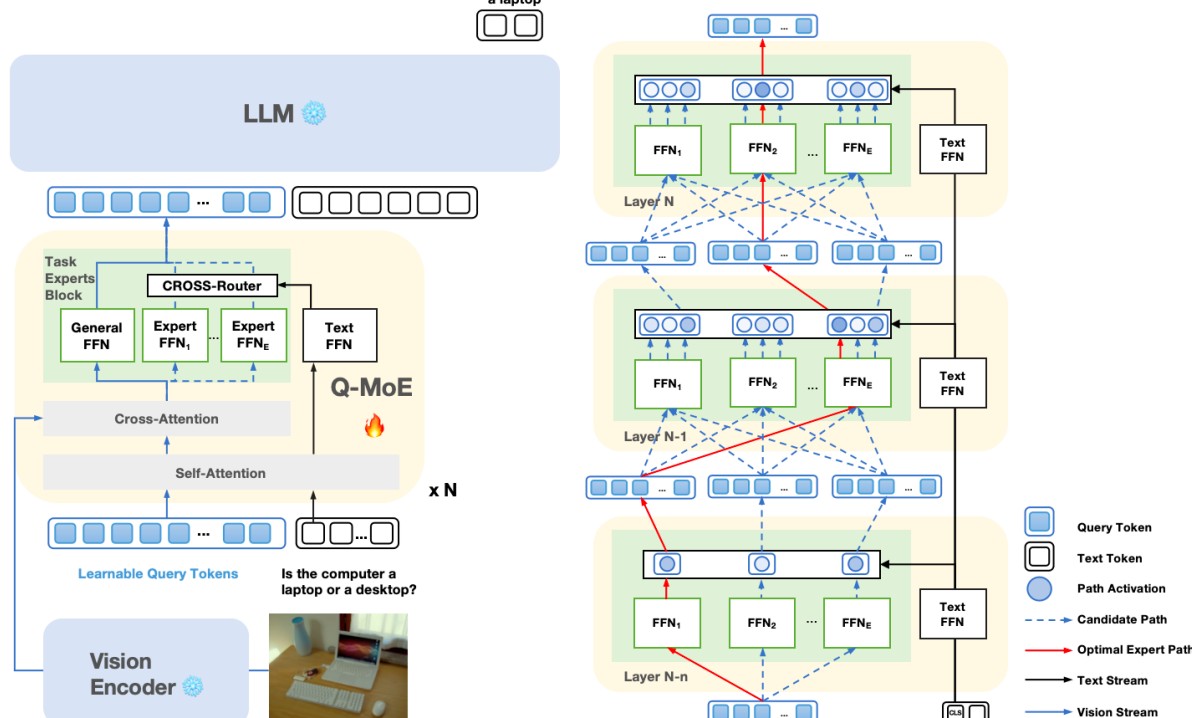

Figure 2: **Architecture of Q-MoE and Optimal expert path based training Strategy.** The **left** side is the architecture of our Q-MoE based MLLMs which consists of a vision encoder, the Q-MoE and a LLM. The **right** side illustrates the optimal expert path based training strategy. We only show the Expert components and their connections in transformers.

LoRA experts is based on the pre-defined clustering result of samples. Although MoCLE incorporates instruction-related information into expert decision-making via clustering, this predetermined clustering limits the flexibility and autonomy of the experts.

These methods typically integrate the MoE architecture into LLMs, overlooking the encoding and transformation on the visual side. Our work incorporates MoE in the connector of the MLLM, instead of in LLM, to utilize different experts to filter more accurate task-specific visual information for the MLLM. Meanwhile, instead of the hard router, we focus on designing a text-driven soft router, Cross-Router, effectively routing query tokens based on sample semantics. Furthermore, we design an optimal expert path based training method to help maximize the effectiveness of the experts combination.

## 3 METHOD

### 3.1 Overview

As shown in **Figure 2**, the overall structure of the MLLM consists of a LLM, a Q-MoE (in yellow box), and a visual encoder. The Q-MoE acts as the connector between LLM and visual encoder.

Q-MoE consists of a Self Attention Block, a Cross Attention Block, a Text FFN, and a Task Experts Block (the green box in yellow box). Same as that in Q-former, Q-MoE employs a set of

learnable query tokens to interact with text and vision in Self Attention Block and Cross Attention Block. The Text FFN encodes text representation. Different from that in Q-former, Q-MoE includes an additional Task Experts Block, where multiple Expert FFNs and a text-driven Cross-Router are designed. With an optimal expert path based training strategy, the Task Experts Block is helpful to route right visual information for the task in hand. We describe Task Experts Block in **Section 3.2** , and the optimal expert path based training strategy in **Section 3.3**.

### 3.2 Task Experts Block

Task Experts Block consists of the Cross-Router, multiple Expert FFNs and the General FFN. We denote $N_q$ the number of query tokens, $E$ the number of Expert FFNs, and $D$ the hidden states dimension. Given a set of query tokens $x \in R^{N_q \times D}$ as input, we product a cross attention between each output of certain Expert FFN and the output of Text FFN in Cross-Router to help decide which Expert FFNs is activated. The General FFN remains continuously activated. Ultimately, the Q-MoE sums the output values of Expert FFNs and the General FFN as the overall output.

Firstly, after interacting with text and vision through attention, query tokens are fed into multiple Expert FFNs. Each FFN is a single Feed Forward Network(Equation (1)), including two weight matrix $W_1$ and $W_2$, an gelu activation, layer normalization (LN)

Anonymous Authors

and residual connection. Simultaneously, the text hidden state is processed through a Text FFN. We denote the output of the $i^{th}$ Expert FFN as $f_i(x) \in \mathbb{R}^{N_q \times D}$ and the output from text FFN as $f_{text}(x) \in \mathbb{R}^{N_q \times D}$.

$$f_i(x) = LN\left(x + \left(W_2\sigma_{gelu}\left(W_1x\right)\right)\right), i \in 1, 2, ..., E \qquad (1)$$

Cross-Router is responsible to assign a probability value to each Expert FFN. In Cross-Router, we take $f_i(x)$ and the hidden state $h_{CLS} \in \mathbb{R}^{1 \times D}$ corresponding to the CLS token in $f_{text}(x)$ as input. Specifically, we treat $h_{CLS}$ as query and $f_i(x)$ defined in Equation (1) as key and value. We use $h_{CLS}$ and $f_i(x)$ to calculate attention scores and weight output for $f_i(x)$ resulting in the output $c_i(x) \in \mathbb{R}^{1 \times D}$, as shown in Equation (2).

$$c_i(x) = \text{softmax}\left(\frac{h_{CLS}f_i(x)^T}{\sqrt{D}}\right)f_i(x) \qquad (2)$$

Then the routing vector $W_g \in \mathbb{R}^{D \times 1}$ to multiply $c_i(x)$. We apply a softmax function to all concatenated $W_g c_i(x)$, and output the activation probability score $g(x)_i$ of $i^{th}$ expert to determine whether and how much it is activated according to Equation (3). The motivation behind this design is to make the decision of FFN activation based on the combination of processing results of the certain FFN and the textual and task-specific information stored in [CLS] token. We also experiment with several kinds of Routers and will further analyze their advantages and disadvantages in **Section 5.2**.

$$g(x)_i = \left(\text{softmax}\left(\left[W_g m(c_0(x)), \dots, W_g m(c_E(x))\right]\right)\right)_i \qquad (3)$$

For a given layer $j$, the output of the $i^{th}$ Expert is $e_i^j(x)$, which is the product of the Expert FFN output $f_i^j(x)$ and the routing probability value $g^j(x)_i$.

$$e_i^j(x) = g^j(x)_i f_i^j(x) \qquad (4)$$

The optimal expert path selection and training strategy will be explained in detail in **Section 3.3**. In this section, we only denote the expert output value on the optimal expert path at the top layer of Q-MoE as $e^{\text{Top}}(x)$.

The General FFN is a parallel structure to the Expert FFNs as shown in **Figure 2**. We expect the Expert FFNs to learn to process information based on task-specific distinctions, while the General FFN learn generic knowledge. Since the General FFN is continuously activated, in Equation (3), we have $i = 1$ and $g(x) = 1$, and define its output in Equation (5).

$$\hat{e}^j(x) = f^j(x), x \in R^{N_q \times D} \qquad (5)$$

Ultimately, the overall output of Q-MoE defined in Equation (6) is the combination of the selected Expert FFN's output and the General FFN's output at the top layer.

$$e_{QMoE}(x) = \hat{e}^{\text{Top}}(x) + e^{\text{Top}}(x) \qquad (6)$$

## 3.3 Optimal Expert Path based Training

Previous Mixture-of-Experts (MoE) normally selects the Top-K(K>=1) experts in each layer. This is essentially a greedy search, where the locally optimal choice of expert is made at each layer. We introduce dynamic program for optimal path based expert path selection as shown in **Figure 2**. The idea is if we consider the FFNs as nodes and the connections between FFNs across layers as edges, different combinations of Expert FFNs in different layers can be seen as an different expert paths. Each path is a task expert. For the given set of query tokens, our goal is to find the optimal path that best leverage task information.

---

**Algorithm 1:** ExpertPath Algorithm

**Input:** ExLayers, E
**Output:** finalPath

1   isFirst <- True
2   **for** $j$ in ExLayers **do**
3     **if** isFirst **then**
4       optimPaths <- (Path($e_1^j$), Path($e_2^j$),...,Path($e_E^j$))
5       isFirst <- False
6       continue
7     **else**
8       candidateExperts <- (Expert($e_1^j$), Expert($e_2^j$),   Expert($e_E^j$));
9       candidatePaths <- None
10      **for** path in optimPaths **do**
11        **for** expert in candidateExperts **do**
12         path.expertList.add(expert)
13         path.pathProb *= expert.expertProb
14         candidatePaths.add(path)
15      optimPaths <- TopK(candidatePaths)
16   return finalPath <- path with highest pathProb in optimPaths;

---

Specifically, assuming Task Experts Blocks are inserted into $n$ transformer layers with $E$ experts per layer, there are a total of $E^n$ expert paths. Each path can be considered a form of encoding for a specific task or sub-task. We name the optimal expert path selection algorithm as *ExpertPath*.

The *ExpertPath* algorithm(**see Algorithm 1**) can be described as follows. Let *ExLayers* be the layers which Task Experts Blocks are inserted, each layer contains $E$ Task Experts. Two structures: *Expert* and *Path* are used. The attribute of the *Expert* is the activation probability *expertProb*, also known as $g(x)_i$ calculated by the router defined in the Equation (3). The attributes of the *Path* include the experts selection list *expertList* through layers and the path overall activation probability *pathProb* which is the production of *expertProb* of all experts in the path.

The algorithm begins with initializing the optimal paths (*optimPaths*) with the first layer's Task Experts as the initial nodes. Furthermore, the current $j^{th}$ layer's experts ($e_1^j, e_2^j, ..., e_E^j$) calculated by Equation (4) are considered as candidate experts (*candidateExperts*). For each expert in the *candidateExperts*, we add it to the *expertList*

| Method | Connector | LLM | GQA test-dev | OKVQA val | VQAv2 val | AOKVOA test | COCOCap val | TextCaps val | Average VQA | Total |
|---|---|---|---|---|---|---|---|---|---|---|
| LLaVA | Projection | LLaMA-13B | 41.3 | 54.4 | - | - | - | - | - | - |
| LLaVA-1.5 | Projection | Vicuna-7B | 62.0* | - | 78.5* | - | - | - | - | - |
| Qwen-VL | CrossAttn | Qwen-7B | 59.3* | 58.6* | 79.5* | - | - | - | - | - |
| MiniGPTv2 | Projection | LLaMA2-7B-chat | 60.3* | 56.9 | - | - | - | - | - | - |
| InstructBLIP | Q-former | Vicuna-7B | 49.2 | 62.1* | - | 75.7* | - | - | - | - |
| InstructBLIP | Q-former | Vicuna-13B | 49.5 | - | - | - | - | 75.6* | - | - |
| QA-ViT | Projection | FlanT5-XXL | - | - | 76.5* | - | 138.2* | 101.7* | - | - |
| BLIP2 | Q-former | FlanT5-XXL | 59.60 | **53.43** | 75.81 | 76.68 | 135.38 | 105.38 | 66.38 | 84.38 |
| **Ours** | **Q-MoE** | FlanT5-XXL | **60.39** | 53.26 | **76.72** | **76.76** | **135.41** | **107.29** | **66.75** | **84.95** |
| BLIP2 | Q-former | Vicuna-7B | 62.46 | 57.52 | 78.18 | **75.20** | 138.60 | 106.61 | 68.34 | 86.43 |
| **Ours** | **Q-MoE** | Vicuna-7B | **63.61** | **58.64** | **79.56** | 75.02 | **139.21** | **108.08** | **69.21** | **87.35** |

Table 1: **Results of fine-tuning under the multi-task setting.** Performance comparison of our Q-MoE based MLLM and some other representative MLLMs with the same size. Notice results with * means the training split of the corresponding dataset is used during tuning.

of each path to form new paths. The new path's *pathProb* is updated by multiplying the *expertProb* of the newly added expert. These new paths forms the *candidatePaths*. Among these candidate, the local *TopK* selection is made, temporarily storing the *TopK* optimal paths sorted by the *pathProb* as input for the next layer. At the top layer, the path with the highest overall activation probability value in *expertProb* is selected as the *finalPath*.

Compared to the greedy selection algorithm of vanilla Mixture-of-Experts , ExpertPath temporarily requires additional storage to store *optimPaths*, including *expertList* of the current layer's *TopK* optimal paths and their corresponding activation values *pathProb*. The *Top*1 selection is made at the very top layer. Since the temporarily stored *TopK* information remains within the current batch, the batch size will increase to $K$ times its original size as shown in **Figure 2**. Since the overall hidden state of Q-MoE is not large and the batch size only increases in the Q-MoE part, there is not a significant impact on the overall memory usage.

In the Task Experts Block, ExpertPath algorithm is used for path selection. At the very top layer, the results of *finalPath* is added to the results of the General FFNs, serving as the overall output according to the Equation (6).

### 3.4 Training Details

We fine-tune the model through the language modeling loss, where the frozen LLM is tasked to generate the text conditioned on the query tokens representation from Q-MoE and the text instructions. Simultaneously, we introduce an importance auxiliary loss to help balance the usage of Task Experts in Q-MoE. The two losses are weighted using an $\alpha$ hyperparameter.

$$\mathcal{L} = \mathcal{L}_{LM} + \alpha * \mathcal{L}_{\text{Imp}} \qquad (7)$$

**Language Modeling loss.** Given the query $x = [x_1, x_2, \cdots, x_{N_q}]$ from Q-MoE and instruction $t = [t_1, t_2, \cdots, t_{N_i}]$ , LLM generates the output sequence $y = [y_1, y_2, \cdots, y_{N_o}]$, where $N_i$ and $N_o$ represents the length of the input and output sequence. The formula is:

$$\mathcal{L}_{LM} = -\sum_{i=1}^{N} \log p(y \mid x, t) \qquad (8)$$

**Importance auxiliary Loss.** The importance of each expert $e_i$ is defined as the sum of normalized probability values for each sample in the batch assigned to $e_i$. Note that $X$ refers the whole batch, $x$ refers to the specific sample, and $g(x)_i$ is defined in **Equation (3)**.

$$\text{Imp}_i(\mathbf{X}) := \sum_{\mathbf{x} \in \mathbf{X}} g(x)_i \qquad (9)$$

The importance auxiliary loss is the squared coefficient of variation of the importance distribution over experts.

$$\mathcal{L}_{\text{Imp}}(\mathbf{X}) = \left( \frac{\text{std}(\text{Imp}(\mathbf{X}))}{\text{mean}(\text{Imp}(\mathbf{X}))} \right)^2 \propto \text{var}(\text{Imp}(\mathbf{X})) \qquad (10)$$

## 4 EXPERIMENTS

We first conduct multi-task fine-tuning experiments to assess the capabilities of Q-MoE along with different LLMs, including Vicuna-7B and FlanT5-XXL. While ensuring that the other components of the model and the fine-tuning datasets remains constant, we compared the performance of fine-tuning Q-former connector and Q-MoE connector based on the architecture of BLIP2. We also evaluate the zero-shot performances of these two connectors after fine-tuning. Finally, we give ablation studies from several different aspects.

### 4.1 Experimental Setup

**Model Settings.** Following BLIP2[21], we utilize EVA-CLIP[36] as the vision encoder. We initialize the model based on checkpoints pretrained on image-text pairs without instruction tuning provided by [5]. The parameter weights of Expert FFNs and General FFN are initialized by the parameter weights of query FFN in Q-former at the specific layer. We insert the Task Experts Block, which consists of 3 Expert FFNs and 1 General FFN, into the top 6 Transformer layers in Q-MoE. As for the optimal expert path based training strategy, we

| Method | LLM | VizWiz Acc | VSR Acc | HM Acc | NoCaps CIDEr |
|--------|-----|------------|---------|--------|--------------|
| Flamingo | 9B | 28.80 | - | 57.00 | - |
| Qwen-VL | Qwen-7B | 35.20 | - | - | 121.40 |
| MiniGPTv2 | LLaMA2-7B-chat | 32.90 | 60.60 | 58.20 | - |
| BLIP2-PT | Vicuna-7B | 25.30 | 50.00 | 50.60 | 107.50 |
| InstructBLIP | Vicuna-7B | 34.50 | 54.30 | 59.60 | 123.10 |
| BLIP2-FT | Vicuna-7B | 31.35 | 60.55 | 52.60 | 103.85 |
| **+Q-MoE** | Vicuna-7B | **31.79** | **61.04** | **53.80** | **104.54** |

Table 2: **Zero-shot performances on the held-out datasets.** VSR and HM denote HatefuleMemes and Visual-Spacial Reasoning. We report accuracy for Vizwiz, VSR and HM, and report the CIDEr score for NoCaps. BLIP2-PT refers to the BLIP2 model after pretraining and text-image alignment without instruction tuning. BLIP2-FT refers to the model fine-tuned under our multi-task setting. The version and setting of Q-MoE correspond with the model listed in **Table 1**. The metrics for other models are from publicly reported papers.

do Top-3($K$ = 3) local optimal selection at each layer and perform Top-1 expert path selection at the top layer. The hyperparameters mentioned above constitute the default settings for the method described in our paper. We freeze the vision encoder and the LLM and only fine-tune the parameters in Q-MoE. We provide additional training details in supplementary.

**Dataset Settings.** We conduct experiments on two major types of tasks, VQA and image captioning. For VQA, we involve GQA[14] for scene understanding and compositional task, OK-VQA[28] for external knowledge based task, VQAv2[12] for general task and A-OKVQA[32] for multi-choice task. For image captioning, we involve COCO caption[25] for general captioning task and TextCaps[35] for scene-text captioning task. For VQA tasks, the text input of Q-MoE is the sample's question and that of LLM is the question with a random selected instruction. For image captioning tasks, we utilize a random instruction to notify the task as the input of both Q-MoE and LLM. The detailed instruction list is provided in supplementary, alongside further details on the training dataset composition and sampling strategy. Overall, our whole training dataset comprises approximately 2M assets. We conduct performance evaluation on the evaluation datasets corresponding to the aforementioned fine-tune tasks and incorporate 4 additional tasks to assess whether the model retains its generalizability after fine-tuning. The zero-shot evaluation datasets include 3 VQA datasets (VSR[26], VizWiz[13] and HM[18]) and 1 caption dataset(NoCaps[1]).

## 4.2 multi-task fine-tuning Experiments

We compare the performances of our Q-MoE based model with some other Representative MLLMs with same size on six different datasets. Mainly, We compare Q-MoE and Q-former while remaining other components and training details the same. As demonstrated in **Table 1**, our Q-MoE structure based MLLM outperforms Q-former based MLLM significantly regardless of whether the LLM is decoder-only model(Vicuna) or encoder-decoder model(FlanT5).

We conduct the average of all VQA evaluation scores in General VQA and average of all tasks in General Total. Under FlanT5-XXL, our Q-MoE gains 0.4 in General VQA and 0.6 in General Total. While under Vicuna 7B, Q-MoE shows improvements of 0.9 and 1.0 respectively. InstructBLIP proves that FlanT5-based model outperforms in multi-choice task and Vicuna-based model in open-ended task, but

Q-MoE mitigates the tendency of over-fitting single task and show balance in multiple tasks through Task Experts Block. By treating information from different tasks differently, Task Experts Block can make Q-MoE learn multiple tasks relatively well, enhancing overall model performance.

Moreover, Q-MoE exceeds QA-ViT by using similar datasets in fine-tuning. With LLM and ViT frozen, by only fine-tuning Q-MoE, we've also achieved comparable results compared to open-source models fine-tuning LLM(LLaVA, Qwen-VL, MiniGPT-v2 and InstructBLIP).

## 4.3 Zero-shot Experiments

In this section, we evaluate the zero-shot performance of our fine-tuned Q-MoE and Q-former. According to **Table 2**, as a connector, Q-MoE demonstrates a certain improvement in zero-shot performance compared to Q-former, suggesting that this connector structure offers enhanced generalizability for held-out datasets. Compared to BLIP2-pretrain, there is an improvement in zero-shot performance on VQA tasks(VSR, VizWiz and HM), though there is a slight decline on the NoCaps dataset. It is worth noting that our method outperforms the same architecture's BLIP2-pretrain and InstructBLIP on the VSR dataset. Moreover, relative to other models with larger amounts of fine-tuning parameters, such as Flamingo, Qwen-VL, and MiniGPTv2, our zero-shot performance remains competitive.

## 5 ABLATION STUDIES

In this section, to verify the effectiveness of Q-MoE, we conduct extensive experiments to understand the performance improvements better and analyze the impact of our method in **Section 5.1**. First, we study the effect of Cross-Router and ExpertPath training strategy which donate the main components of Q-MoE. Then in **Section 5.2**, we compare the differences of designed routing functions. Next, we analyze the numbers and combination of experts **Section 5.3**. Furthermore, we demonstrate the adaptability of the Q-MoE structure to multiple task combinations by increasing the number of tasks in **Section 5.4**.

Due to space constraints, in the ablation studies, we report the metrics for VQA and Caption tasks by averaging their respective indicators. Specifically, "VQA" represents the average evaluation score

| Components | VQA | Caption | Average | Drop |
|---|---|---|---|---|
| Q-MoE | **69.21** | **123.65** | **87.35** | |
| - CROSS | 68.98 | 123.57 | 87.17 | -0.18 |
| - ExpertPath | 68.73 | 123.24 | 86.90 | -0.27 |

Table 3: **Ablation of Main Components.** We separately replace Cross-Router and ExpertPath from Q-MoE. Default settings are marked in gray.

| routing | Strategy | VQA | Caption | Average |
|---|---|---|---|---|
| - | - | 68.34 | 122.60 | 86.43 |
| PRE | | 68.29 | 122.64 | 86.41 |
| CLS | Path | 68.50 | 121.81 | 86.27 |
| POST | Optim | 68.31 | 123.14 | 86.59 |
| CROSS | | **68.60** | **124.15** | **87.12** |

Table 4: **Abaltion of routing.** Compare different routing functions, including PRE, CLS, POST, CROSS. We experiments with the Q-MoE with 4 Task Experts and with the ExpertPath training strategy. First line represents the results of 1 expert without routing and strategy. Model utilizing Cross-Router is highlighted in grey.

across vqa-related datasets (GQA, VQAv2, OKVQA, AOKVQA), while "Caption" denotes the average evaluation score for caption-related datasets (COCOCaption, TextCaption). For detailed VQA and Image Captioning evaluation scores, please refer to the **Appendix** in the supplementary.

## 5.1 Main Components

To investigate the effect of our core components, Cross-Router and ExpertPath training strategy, we observed the overall impact by removing these two components separately according to **Table 3**. The first row presents the original setup of our Q-MoE.

The second row shows the replacement of Cross-Router with POST-Router, which omits the CLS guided semantic insertion. It is evident that there is a performance decline in both VQA and Caption tasks, indicating that incorporating CLS hidden state-related semantic information into routing is necessary.

The third row describes the removal of the ExpertPath training strategy from Q-MoE, by reverting to the original vanilla MoE approach. Specifically, we involve a greedy search for experts at each layer, making single-layer decisions for the top1 expert. The results show that ExpertPath significantly impacts the final performance. This path search strategy more effectively identifies the optimal combination of experts, surpassing the original greedy search method in effectiveness.

## 5.2 Routing Functions

Determining how the inputs are routed to different experts. To explore the effects of different routing styles, we conduct several experiments as follows. We set the number of Expert FFNs in 4 and remove the General FFN. And we compare different activation strategies, including PRE, POST, CLS and CROSS. PRE refers to the vanilla MoE routing function which determines the expert

| #Ex | #GenEx | #Param | VQA | Caption | Average |
|---|---|---|---|---|---|
| 1 | | 188M | 68.34 | 122.60 | 86.43 |
| 3 | | 245M | 69.03 | 123.48 | 87.18 |
| 2 | 1 | 245M | 68.71 | **124.17** | 87.19 |
| 4 | | 273M | 68.60 | 124.15 | 87.12 |
| 3 | 1 | 273M | **69.21** | 123.65 | **87.35** |
| 7 | 1 | 387M | 68.77 | 123.96 | 87.16 |

Table 5: **Ablation of Expert Setup and Combination.** #Ex denotes the number of Expert FFNs and #GenEx denotes the number of General FFNs. #Param means how many parameters are activated during training. Default setting is in grey.

| Tasks | VQA | AOKVQA | COCOCap | TextCaps | Average |
|---|---|---|---|---|---|
| VQA | 65.67 | | | | 65.67 |
| | 67.31 | | | | **67.31(+1.64)** |
| +AOKVQA | 66.83 | 72.49 | | | 68.24 |
| | 67.08 | 74.32 | | | **68.89(+0.65)** |
| +COCOCap | 67.21 | 73.10 | 139.77 | | 82.90 |
| | 67.30 | 74.93 | 139.14 | | **82.99(+0.09)** |
| +TextCaps | 66.92 | 75.20 | 138.60 | 106.61 | 86.43 |
| | 67.27 | 75.02 | 139.21 | 108.08 | **87.35(+0.94)** |

Table 6: **Ablation of Training Tasks.** As the training dataset expands, we sequentially report the metrics for the evaluation set of the fine-tuned datasets. "Average" refers to the average of the evaluation scores across the datasets that participated in the assessment.

selection before FFN. As illustrated in Equation (2), the Cross-Router operates on the cross attention result of the FFN's output and [CLS] token hidden states. POST and CLS Router only operates on the FFN's output and [CLS] token hidden states respectively. Detailed equations of PRE, POST and CLS Router are in Appendix.

As shown in **Table 4**, Cross-Router which fuses the information from FFN and sample semantic outperforms other Routers. Therefore, using the CLS hidden state as a guide for sample semantics during the routing process effectively directs the selection of the FFN.

## 5.3 Experts Setup and Combination

We differentiate tasks in different extent by setting the number and combination of experts. As demonstrated in **Table 5**, the increase in the number of experts does not give rise to an increase in overall performance. With a small number of experts, there is no high requirement of task differentiation. While with a large number of experts, there is a high requirement for routing, however, not necessarily leading to performance improvement. With the joining of general expert, we can regard the learning of task expert as incremental learning, where general expert learns fundamental knowledge and Task Expert learns incremental knowledge. Combining general expert with multiple Task Experts is effective, and among all it's best to set 3 Task Experts and 1 general expert.

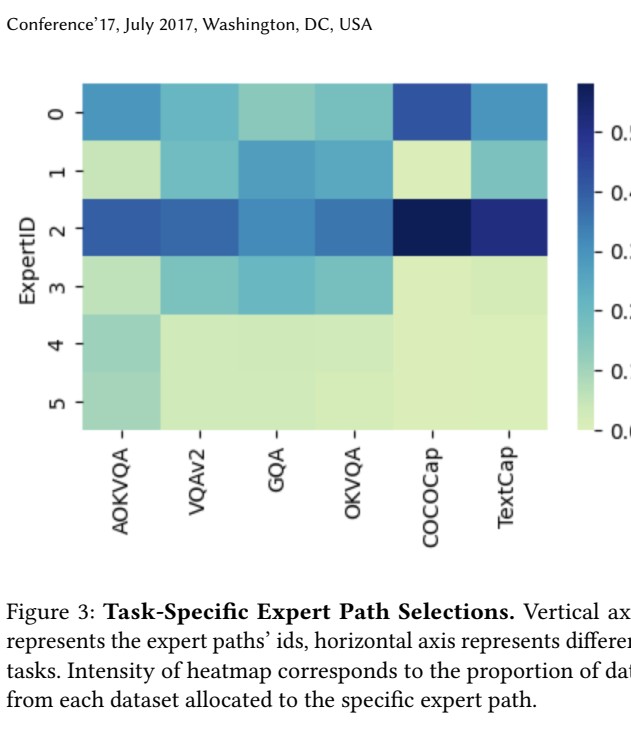

Figure 3: **Task-Specific Expert Path Selections.** Vertical axis represents the expert paths' ids, horizontal axis represents different tasks. Intensity of heatmap corresponds to the proportion of data from each dataset allocated to the specific expert path.

## 5.4 Effects on multiple tasks

Among the six fine-tuning datasets we selected, VQA, GQA, and VQAv2 can be collectively considered as generative VQA tasks, while the sample input spaces for AOKVQA(multi-choice VQA), COCOCaption(General captioning), and TextCaption(text-based captioning) are slightly different from the former ones. We gradually involve these tasks and examine whether Q-MoE can maintain stable performance across different numbers of tasks in **Table 6**.

As we can see, Q-MoE out-performs Q-former in Generative VQA task by 1.64 on General VQA metric. Additionally incorporate Multi-Choice VQA, Q-MoE still gains 0.65 performance increase and shows significant improvement in A-OKVQA. We believe that this is because the task forms of Multi-Choice VQA and Open-Ended VQA are slightly different, and we are able to bring performance gains by adding a Task-specific processing structure that treats the two subtasks differently. Next, after introducing Image Captioning Task, Q-MoE still outperforms Q-former by 0.09 and 0.94.

## 6 TASK-SPECIFIC EXPERT PATHS

To better capture how experts react to different tasks, we make visualization of the expert paths on all datasets, using their evaluation set. As the heat-map shown in **Figure 3**, under the setting of 3 task experts and 1 general task, we select top-6 expert paths as shown in the vertical axis. The darker block in heat-map, the more frequently task is routed into this expert path. It is evident that samples from COCO caption and Text caption datasets are primarily assigned to expert path 0 and 2. The VQAv2 dataset predominantly utilizes expert path 2 and 3, while the GQA and OK-VQA tasks mainly involve expert path 1 and 2. Despite being a VQA dataset, AOKVQA presents a multi-choice task, which is slightly different from generative VQA dataset and shows preference for expert path 4 and 5.

Ultimately, it shows different expert path preference between VQA tasks and captioning tasks. Q-MoE could tell the slight difference in the distribution of tasks, proving the effectiveness of our expert routing mechanism.

## 7 DISCUSSION AND CONCLUSIONS

In this work, we introduced Q-MoE, an innovative connector to query task-specific visual information utilizing a Mixture of task experts. We design

the Cross-Router to leverage sample semantics for guiding the selection of task experts effectively. Meanwhile, we conceptualize the combination of experts as expert paths, and design the ExpertPath training strategy. This strategy aims to identify the optimal expert path, shifting from local optimization to global optimization. Through extensive experimentation, we have demonstrated the effectiveness of our method. It enhances the performance of MLLMs under the multi-task setting.

Q-MoE aims to differentiate information within the connector in a task-specific manner. Our findings highlight the pivotal role of designing a Router capable of distinguishing between different tasks, and designing the strategy to find the optimal expert collaboration. This paper presents Cross-Router and ExpertPath as solutions to these critical points. Furthermore, we are focusing on extending the application of Cross-Router and ExpertPath to other components of MLLMS, broadening the utility of our approach. We hope our method will inspire further research focusing on efficient expert routing and more optimal expert training strategies.

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
