# OpenReview forum: "Q-MoE: Connector for MLLMs with Text-Driven Routing"
_acmmm.org/ACMMM/2024/Conference — MM2024 Poster_

### Official Review · Reviewer_N8K7 · 2024-05-22

**Rating:** 4
**Confidence:** 3

**Summary:**

This paper proposes Q-MoE, a query-based connector with Mixture-of-Experts
(MoE) to extract task-specific information with text-driven routing. Furthermore, an optimal path based training strategy is also
proposed to find an optimal expert combination. Extensive experiments on two popular open-source LLMs and several different
visual-language tasks demonstrate the effectiveness of the Q-MoE connecter.

**Strengths:**

a) The experiment in this paper is thorough, and the presentation is relatively clear.

b) The motivation of the paper is reasonable. Most of the previous connectors, such as light-weight projection and Q-former, treat visual information for diverse tasks uniformly, therefore lacking task-specific visual information extraction capabilities. So this paper proposes Q-MoE, a query-based connector with Mixture-of-Experts (MoE) to extract task-specific information with text-driven routing.

c) Experiments conducted under multi-task settings as well as zero-shot setting demonstrate the remarkable proficiency of the proposed model in various vision language tasks.

**Limitations:**

a) Figure 1 is overly simplistic and does not indicate the flow of data.

b) What does 'm' represent in Formula 3? This is confusing.

c) The proposed ExpertPath Algorithm is somewhat similar with beam search, I am worry about the complexity during inference , can you report the time compared with the baseline model in Table 4.

d) The model proposed in this paper activates all experts, which appears to consume more memory and compute compared with baseline model. As contrast, Sparse MoE is much more fancy. A comparison between the proposed Q-MoE and the sparse MoE model is needed in terms of memory consumption and performance.

**Suitability:**

3

---

### Official Review · Reviewer_zjTc · 2024-05-27

**Rating:** 4
**Confidence:** 2

**Summary:**

The paper introduces a novel query-based connector for Multimodal Large Language Models (MLLMs), addressing the limitations of existing connectors by utilizing a Mixture-of-Experts (MoE) architecture with text-driven routing for task-specific visual information extraction. The proposed optimal path-based training strategy enhances the model's performance by selecting the best combination of experts across layers. Extensive experiments demonstrate significant improvements in various vision-language tasks, including Visual Question Answering (VQA) and image captioning, compared to previous methods. Comprehensive ablation studies highlight the contributions of different components, and visualization of expert path selections indicates effective task differentiation. The results showcase Q-MoE's potential for broader applications and inspire future research in efficient expert routing and training strategies.

**Strengths:**

* The paper introduces Q-MoE, a novel query-based connector that leverages a Mixture-of-Experts (MoE) architecture to extract task-specific visual information. This approach addresses the limitations of previous connectors which treated visual information uniformly across diverse tasks.
* The incorporation of a text-driven routing mechanism within the MoE framework is a novel concept, enabling more accurate and context-specific visual information extraction. The optimal path-based training strategy further enhances the theoretical framework by shifting from local to global optimization in expert selection.

**Limitations:**

* The optimal path-based training strategy and the use of multiple experts can be resource-intensive, requiring substantial computational power and memory. The author might need to consider to use experiments to demonstrate the extra computation overhead of such methods.
* I am wondering whether such Q-MoE can be extended to apply to other tasks and achieve good performance?

**Suitability:**

3

---

### Official Review · Reviewer_coNu · 2024-05-29

**Rating:** 4
**Confidence:** 3

**Summary:**

The paper introduces a novel connector structure named Q-MoE to enhance Multimodal Large Language Models for vision-language tasks. Q-MoE utilizes a text-driven routing mechanism to dynamically select task-specific expert networks, thereby improving the extraction of relevant visual information. Additionally, the paper proposes the ExpertPath training strategy, which optimizes the combination of these experts across different layers, leading to significant performance improvements in tasks such as visual question answering and image captioning. Through extensive experimentation, the Q-MoE connector demonstrates its effectiveness in diverse vision-language scenarios.

**Strengths:**

The paper introduces the Q-MoE connector that combines Mixture-of-Experts with text-driven routing to enhance task-specific visual information extraction for vision-language tasks. The introduction of an optimal expert path based training strategy, ExpertPath, shifts from local optimization to global optimization of experts. This approach maximizes the effectiveness of expert combination, leading to significant improvements in various vision-language tasks under multi-task and zero-shot settings.
The effectiveness of the Q-MoE structure is validated through extensive experiments on various vision-language tasks, including Visual Question Answering, knowledge-based VQA, image captioning, and text-based image captioning.
Additionally, the paper is well-written and clearly presented, making it accessible to readers.

**Limitations:**

1. The introduction of the Q-MoE connector, particularly the ExpertPath strategy, may increase the overall complexity of the model. Detailed comparisons of the time and space costs should be included in the paper.
2. The Q-MoE structure employs three expert FFNs and one general FFN, but the ablation study in Table 4 uses four expert FFNs. The paper lacks an explanation for why the ablation study was conducted with a different structure than the one adopted by the model.
3. The paper mentions that previous MoE models normally select the Top-K(K>=1) experts in each layer, while the proposed method uses only the Top1 path. It's better to explain the motivation behind replacing TopK with Top1.

**Suitability:**

3

---

### Meta-Review · Area_Chair_ikt3 · 2024-07-01

**Recommendation:** Accept (Poster)
**Confidence:** 3

**Metareview:**

This paper proposes Q-MoE, a query-based connector with Mixture-of-Experts (MoE) for visual-language tasks. It is novel and effective enough for its acceptance. All three reviewers are positive.